# Susceptibility of Ovine Bone Marrow-Derived Mesenchymal Stem Cell Spheroids to Scrapie Prion Infection

**DOI:** 10.3390/ani13061043

**Published:** 2023-03-13

**Authors:** Adelaida Hernaiz, Paula Cobeta, Belén Marín, Francisco José Vázquez, Juan José Badiola, Pilar Zaragoza, Rosa Bolea, Inmaculada Martín-Burriel

**Affiliations:** 1Laboratorio de Genética Bioquímica (LAGENBIO), Facultad de Veterinaria, Universidad de Zaragoza, IA2, IIS-Aragón, 50013 Zaragoza, Spain; 2Centro de Encefalopatías y Enfermedades Transmisibles Emergentes (CEETE), Facultad de Veterinaria, Universidad de Zaragoza, IA2, IIS-Aragón, 50013 Zaragoza, Spain; 3Equine Surgery and Medicine Service, Veterinary Hospital (HVUZ), Universidad de Zaragoza, 50013 Zaragoza, Spain; 4Departamento de Patología Animal, Facultad de Veterinaria, Universidad de Zaragoza, 50013 Zaragoza, Spain; 5Centro de Investigación Biomédica en Red de Enfermedades Neurodegenerativas (CIBERNED), Instituto de Salud Carlos III, 28029 Madrid, Spain

**Keywords:** 3D culture, spheroids, mesenchymal stem cells, scrapie, prion

## Abstract

**Simple Summary:**

Prion diseases are fatal and incurable neurodegenerative disorders affecting both humans and animals. The development of in vitro cellular models from naturally susceptible species such as ruminants can be very useful for the study of prion disease mechanisms and the discovery of potential therapies. Our study shows for the first time how the culture, in the form of three-dimensional spheroids of ovine mesenchymal stem cells derived from bone marrow in growth and neurogenic conditions, makes these cells more permissive to prion infection, mimicking the prion toxicity occurring in these diseases. This three-dimensional system appears as a potential in vitro model for studying prion diseases in a microenvironment approaching in vivo conditions.

**Abstract:**

In neurodegenerative diseases, including prion diseases, cellular in vitro models appear as fundamental tools for the study of pathogenic mechanisms and potential therapeutic compounds. Two-dimensional (2D) monolayer cell culture systems are the most used cell-based assays, but these platforms are not able to reproduce the microenvironment of in vivo cells. This limitation can be surpassed using three-dimensional (3D) culture systems such as spheroids that more effectively mimic in vivo cell interactions. Herein, we evaluated the effect of scrapie prion infection in monolayer-cultured ovine bone marrow-derived mesenchymal stem cells (oBM-MSCs) and oBM-MSC-derived spheroids in growth and neurogenic conditions, analyzing their cell viability and their ability to maintain prion infection. An MTT assay was performed in oBM-MSCs and spheroids subjected to three conditions: inoculated with brain homogenate from scrapie-infected sheep, inoculated with brain homogenate from healthy sheep, and non-inoculated controls. The 3D conditions improved the cell viability in most cases, although in scrapie-infected spheroids in growth conditions, a decrease in cell viability was observed. The levels of pathological prion protein (PrP^Sc^) in scrapie-infected oBM-MSCs and spheroids were measured by ELISA. In neurogenic conditions, monolayer cells and spheroids maintained the levels of PrP^Sc^ over time. In growth conditions, however, oBM-MSCs showed decreasing levels of PrP^Sc^ throughout time, whereas spheroids were able to maintain stable PrP^Sc^ levels. The presence of PrP^Sc^ in spheroids was also confirmed by immunocytochemistry. Altogether, these results show that a 3D culture microenvironment improves the permissiveness of oBM-MSCs to scrapie infection in growth conditions and maintains the infection ability in neurogenic conditions, making this model of potential use for prion studies.

## 1. Introduction

Cell cultures have proven to be useful tools for a variety of applications in research. There are currently many established cell lines and primary cultures available and widely used for different purposes. The two-dimensional (2D) platforms in which flat monolayer cells are cultured are most commonly used for research in cell-based assays [1]. These 2D cell culture systems are accessible, convenient, and cost-effective [1]. However, various limitations are still of concern, including the failure to imitate the in vivo architecture and microenvironments. Compared with in vivo cells, 2D-cultured cells possess many different features, such as morphological characteristics, proliferation and differentiation potentials, cell–cell and cell-surrounding matrix interactions, and signal transduction [1,2,3].

To overcome these drawbacks, three-dimensional (3D) cell culture platforms have emerged as a promising approach. Although the optimal 3D condition requirements vary between cell types, and the characteristic features of cells in 3D cultures differ in accordance with their types, these 3D culture systems have proven to be more realistic for translating cell-based assay findings to in vivo applications due to their ability to closely mimic the behavior of in vivo cells [4]. There are different 3D culture systems, including spheroids, organoids, hydrogel embedding, bioreactors, scaffolds, and bioprinting [1,5], with potential applications in drug discovery, disease modeling, and tissue engineering [6,7,8].

Spheroids are spherical-shaped multicellular aggregates that can be formed from several types of cells, including mesenchymal stem cells (MSCs) [9]. They are able to mimic cell–cell and cell–matrix interactions more effectively than 2D cultures, but they lack the capacity to recapitulate the tissue organization exhibited in vivo [10]. In MSCs, spheroid formation enhances the characteristics of these cells by improving their stemness, facilitating the differentiation to multiple lineages, and delaying the in vitro replicative senescent processes [11,12,13].

In the field of neurodegenerative diseases, 3D cell culture platforms appear as good models to reproduce different features of neurodegeneration linked to these diseases. In Alzheimer’s disease (AD), 3D hydrogel-embedded human neural stem cells [14,15], neural progenitor cells [16,17], and scaffold-encapsulated induced pluripotent stem cell (iPSC)-derived neural progenitor cells [18] can recapitulate amyloid-beta aggregation and accumulation of hyperphosphorylated tau, which are key hallmarks of this disease. Furthermore, a triculture model using neurons, astrocytes, and microglia, constructed in a microfluidic platform, also showed these AD hallmarks and allowed the study of microglia recruitment, neuroinflammatory response, and neuron/astrocyte damage [19]. Human iPSC-derived hippocampal spheroids [20] and brain organoids [21,22,23] have been developed, which also have the ability to mimic AD’s pathology and the potential to be used for screening therapeutic strategies. In Parkinson’s disease (PD), dopaminergic neurons [24,25] and SH-SY5Y neuroblastoma cells [26,27] cultured in matrigel-based platforms can model several PD features, such as α-synuclein accumulation and Lewy body-like inclusions. Midbrain organoids carrying the *LRRK2* G2019S mutation have also been created to successfully model *LRRK2*-associated sporadic PD [28].

Prion diseases, or transmissible spongiform encephalopathies (TSEs), are neurodegenerative disorders caused by a pathological misfolded protein derived from an innocuous cellular prion protein (PrP^C^) called PrP^Sc^ [29]. These diseases occur in humans and animals, and among the various types of TSEs, the one affecting sheep and goats, known as scrapie, was the first to be discovered. It is considered a good model for studying different disease aspects in these pathologies [30,31,32]. Regarding 3D culture methods, only two studies using human cerebral organoids have been reported. In these studies, iPSC-derived human cerebral organoids were able to uptake and propagate sporadic Creutzfeldt–Jakob disease (sCJD) prions [33], and also responded to an anti-prion compound by showing delayed prion propagation [34].

A previous study conducted by our group evaluated the effect of scrapie prion infection in ovine bone marrow-derived mesenchymal stem cells (oBM-MSCs) cultured in growth conditions and subjected to neurogenic differentiation [35]. In this study, oBM-MSCs initially took up PrP^Sc^, but they were not able to maintain it over time. MSC-derived neuron-like cells, on the contrary, absorbed and maintained stable PrP^Sc^ levels throughout the duration of the culture [35]. These results were obtained using a 2D culture system approach that may not exactly mimic the physiological response of these cells to prion infection. Therefore, in the current study, we decided to assess the effect of scrapie prion infection in oBM-MSCs-derived spheroids cultured in growth and neurogenic conditions, analyzing their ability to maintain prion infection and the impact of this infection on cell viability.

## 2. Materials and Methods

### 2.1. Bone Marrow Extraction and Ovine Mesenchymal Stem Cell Isolation and Culture

A bone marrow sample was obtained from an adult female rasa aragonesa sheep of two years of age who carried the ARQ/ARQ genotype for the *PRNP* gene. This sheep belonged to a flock from the Center of Encephalopathies and Emerging Transmissible Diseases (CEETE; University of Zaragoza), maintained for research purposes. After the sedation with xylazine and the local anesthesia with lidocaine, bone marrow aspirate was collected from the humeral head as previously described [36]. All procedures were approved by the Ethical Committee for Animal Experiments from the University of Zaragoza (project license PI44/18) and were in accordance with the Spanish Policy for Animal Protection, RD53/2013, and the European Union Directive 2010/63.

MSC isolation from the bone marrow aspirate (3 mL) was performed following the previously described protocol [36,37,38]. This protocol is based on the separation of the mononuclear fraction after density gradient centrifugation in Lymphoprep (Atom) and further isolation thanks to the ability of MSCs to adhere to plastic. After isolation, cells were expanded up to passage 2 in a basal medium consisting of low glucose Dulbecco’s modified Eagle’s medium (DMEM, Sigma-Aldrich, St. Louis, MO, USA), supplemented with 10% fetal bovine serum (FBS), 1% L-glutamine (Sigma-Aldrich), and 1% streptomycin/penicillin (Sigma-Aldrich).

All the subsequent experiments were performed using this oBM-MSC culture. The number of technical replicates used in each experiment is shown in Appendix A.

### 2.2. Ovine Mesenchymal Stem Cell Characterization

The minimal criteria to characterize MSCs are their plastic-adherence capacity in standard culture conditions and their ability to differentiate to mesodermal lineages (adipocytes, osteoblasts, and chondroblasts) in vitro [39].

Adipogenic, osteogenic, and chondrogenic differentiation was evaluated in vitro. The differentiation into mesodermal lineages was performed using specific commercial differentiation kits for osteogenic (StemPro^®^ Osteogenesis Differentiation Kit, Gibco, Life Technologies, Waltham, MA, USA) and chondrogenic (StemPro^®^ Chondrogenesis Differentiation Kit, Gibco, Life Technologies) differentiation. For adipogenic differentiation, basal medium supplemented with 1 μM dexamethasone, 500 μm 3-Isobutyl-1-methylxanthine (IBMX), 200 μM indomethacin, and 15% of rabbit serum was used [38,40]. Subsequently, adipogenic differentiation was confirmed by 0.3% Oil Red O (Sigma-Aldrich) staining after 8 days of differentiation; Alcian Blue G dye (1: 1 in methanol) (Sigma-Aldrich) was used to confirm chondrogenic differentiation at day 15; and osteogenesis was verified with 2% Alizarin Red S staining (Sigma-Aldrich) after 21 differentiation days.

### 2.3. Formation and Culture of Ovine Mesenchymal Stem Cell Spheroids

For spheroid formation, 96-well Nuclon Sphera low-attachment plates (ThermoFisher Scientific, Waltham, MA, USA) were used. In each well, 45,000 cells were seeded to form a unique spheroid per well. The cells were seeded in droplets, and afterwards, 200 μL of basal medium was added into each well. To ensure a correct formation of the spheroids, the plates were incubated for 5 days at 37 °C with 5% CO_2_ without changing the medium. Afterwards, the medium was changed every 48–72 h. Seven days after formation, the spheroids were stabilized and ready to perform the subsequent assays.

### 2.4. Neurogenic Differentiation of Ovine Mesenchymal Stem Cells and Spheroids

#### 2.4.1. Neurogenic Differentiation

Ovine BM-MSCs were seeded at 5000 cells/cm^2^ and differentiated into neuron-like cells using a HyClone AdvanceSTEM^TM^ Neural Differentiation Kit (ThermoFisher Scientific). After 24 h of culture under neurogenic conditions, cells showed a neuron-like morphology, which was monitored by optical microscopy. The pick of neurogenic differentiation was observed at 72 h, as previously described [36]. To maintain the cells in the differentiation state, the medium was changed every 48 h.

For the neurogenic differentiation of spheroids, after they were stabilized, the same neural induction kit was used as that for the 2D cultures, following the same guidelines for medium change.

#### 2.4.2. Nissl Bodies Staining

To verify the correct neurogenic differentiation, neuron-like cells and spheroids were stained with 1% Cresyl Violet Solution (abcam) to detect Nissl bodies, which are granular structures present in neuronal cell bodies and composed of rough endoplasmic reticula rich in RNA [41]. The staining was performed after 3 days of neurogenic differentiation.

#### 2.4.3. Expression Analysis of Neuronal Markers

The expression of a set of neuronal markers was assessed by real-time quantitative PCR (RT-qPCR) in both neuron-like cells and spheroids. These markers were *NEFM* (neurofilament medium chain), *MAP2* (microtubule-associated protein 2) and *TUBB3* (tubulin beta 3 class III) [37]. *HPRT* (hypoxanthine phosphoribosyltransferase) and *G6PD* (glucose-6-phosphate dehydrogenase) were used as housekeeping genes [37].

For total RNA extraction from 2D cell cultures and further retrotranscription to cDNA, the Cells-to-cDNA^TM^ II kit (ThermoFisher Scientific) was used. RNA from each of the two technical replicates was retrieved from two wells of a P6 culture plate.

Total RNA was extracted from 42 pooled spheroids using the Direct-zol^TM^ RNA Miniprep kit (Zymo Research, San Diego, CA, USA). Two technical replicates were analyzed. The quality and quantity of these RNA samples were checked with a NanoDrop spectrophotometer (Thermo Fisher Scientific, Waltham, MA, USA), and a Qubit 4.0 fluorometer (Life Technologies, Carlsbad, CA, USA). cDNA was then obtained using the qScript^TM^ cDNA SuperMix (QuantaBio). All procedures were performed following the manufacturer’s instructions.

Gene expression was quantified by RT-qPCR using the Fast SYBR^TM^ Green Master Mix (Applied Biosystems, ThermoFisher Scientific, Waltham, MA, USA) in a QuantStudio 3 Real-Time PCR instrument (Applied Biosystems). A dissociation curve was performed after every RT-qPCR reaction to confirm the amplification of a single amplicon. The primer design in different amplicons avoided genomic DNA amplification. The comparative quantification of the results was standardized by the 2^−∆∆Ct^ method [42], using the geometric mean of the *HPRT* and *G6PD* housekeeping genes as a normalizer.

### 2.5. Scrapie Inocula and Infection of 2D Ovine Mesenchymal Stem Cell Cultures and Spheroids in Growth and Neurogenic Conditions

Two different inocula were created using central nervous system (CNS) samples: one from a healthy sheep (negative control) and one from a classical scrapie-infected sheep, both carrying the ARQ/ARQ genotype. The inocula were preserved at the tissue bank of the CEETE, University of Zaragoza. The presence/absence of PrP^Sc^ in the CNS tissues was confirmed with two rapid diagnostic tests (Prionics-CheckWestern blotting; ThermoFisher Scientific and Idexx HerdChek; IDEXX, Westbrook, ME, USA) as previously descried [43]. CNS homogenates and microbe safety tests were prepared as described in previous works [35].

The effect of prion infections and their ability to replicate prions were investigated in 2D oBM-MSC cultures and spheroids in growth and neurogenic conditions. Ovine BM-MSCs were seeded at 5000 cells/cm^2^, and each spheroid was generated from 45,000 oBM-MSCs. For each condition, three groups were established: positive (infected with scrapie inoculum), negative (with inoculum from healthy sheep), and control (without inocula). For infection, basal medium was substituted by inocula diluted 1:10 in DMEM medium (10% FBS, 1% L-glutamine and 1% streptomycin/penicillin) for the 2D oBM-MSCs and spheroids in growth conditions, and in HyClone medium for the 2D oBM-MSCs and spheroids subjected to neurogenic differentiation. Cells and spheroids were maintained in this medium for 48 h to analyze cell viability and prion replication ability. Afterwards, the medium was changed twice a week.

### 2.6. PrP^Sc^ Detection

#### 2.6.1. ELISA (Enzyme-linked Immunosorbent Assay)

To test the ability to replicate PrP^Sc^ of 2D oBM-MSCs and spheroids inoculated with scrapie-infected brain homogenate in growth and neurogenic conditions, the amount of pathogenic protein was quantified by ELISA at 2, 5, and 8 dpi using the EEB-Scrapie HerdCheck kit (IDEXX) and following the manufacturer’s recommendations. A previous study conducted by our group showed that this kit was suitable for the sensitive detection of PrP^Sc^ in oBM-MSC cultures [35]. oBM-MSCs were seeded at 5000 cells/cm^2^ in 6-well plates, and the retrieval of the cells was performed by means of trypsinization and subsequent centrifugation to obtain 3 technical replicates per condition and time (dpi). Spheroids formed from 45,000 cells were also collected, and 3 spheroids/technical replicates were analyzed per condition and time (dpi).

#### 2.6.2. Immunocytochemistry of Infected Spheroids

The presence of PrP^Sc^ in spheroids was also confirmed by immunocytochemistry at 5 dpi. Spheroids in growth and neurogenic conditions were fixed in 4% paraformaldehyde and stained with 0.2% eosin. Each spheroid was then included in a 1% agarose matrix; subsequently, the matrices containing the spheroids were embedded in paraffin. Afterwards, paraffin-embedded spheroids were cut to obtain 3-micrometer-thick slices. Hematoxylin–eosin staining was performed to evaluate the integrity of the spheroids after their inclusion in paraffin.

For immunocytochemistry, after deparaffination and rehydration, spheroid sections were digested with 4 µg/mL of proteinase K for 15 min at 37 °C, then subjected to antigen retrieval with citrate buffer (pH 6.0) for 10 min at 96 °C in a PTLink (Dako). Afterwards, endogenous peroxidase activity was blocked using a precast solution (Dako Agilent, Glostrup, Denmark). Sections were then incubated for 30 min at room temperature with the primary antibodies L42 (1:500, R-Biopharm, Darmstadt, Germany) and F89 (1:2000, R-Biopharm, Darmstadt, Germany). Omission of the primary antibody served as a background control for nonspecific binding of the secondary antibody. Subsequently, the sections were incubated with an anti-mouse enzyme-conjugated EnVision polymer (Dako Agilent, Glostrup, Denmark) for 30 min at room temperature. Diaminobenzidine (DAB, Dako Agilent, Glostrup, Denmark) was used as the chromogen. The spheroid sections were assessed and photographed using a Zeiss Axioskop 40 optical microscope (Zeiss, Jena, Germany) and AxioVision Rel.4.7 software.

### 2.7. Cell Viability Assay

An MTT assay was performed to evaluate cell viability and early prion toxicity in oBM-MSCs and neuron-like cells at 2, 5, and 8 days post-inoculation (dpi). Inoculum removal was performed at 2 dpi considering the day of infection with the inocula as day 0. Cell viability in oBM-MSCs and neuron-like cells was studied in positive, negative, and control cultures at 2, 5, and 8 dpi, coinciding with the day of inoculum removal, 2 dpi. Cells were seeded in 96-well plates at 5000 cells/cm^2^, and 8 technical replicates were analyzed for each condition and time (dpi). The MTT assay was carried out as previously described [35].

The viability after prion infection was also analyzed in spheroids in both growth and neurogenic conditions by MTT assay at 2, 5, and 8 dpi in 3 conditions (positive, negative, and controls). Spheroids were formed from 45,000 cells per spheroid in 96-well Nuclon Sphera low-attachment plates (ThermoFisher Scientific), and 4 technical replicates were analyzed for each condition and each time point (dpi). An MTT assay was performed using the same reagents as in oBM-MSCs and neuron-like cells, but the incubation times were different: spheroids were incubated at 37 °C for 24 h with MTT solution, and for 4 h at room temperature while protected from light with HCl solution [44].

In both cases, differences in cell viability were assessed with Student’s *t*-test, defining *p* < 0.05 as a statistically significant difference.

## 3. Results

### 3.1. Ovine Mesenchymal Stem Cell Differentiation into Mesodermal Lineages

To characterize oBM-MSCs, their ability to differentiate into adipocytes, chondrocytes, and osteocytes was assessed.

Cells under adipogenic conditions presented red cytoplasmatic lipid droplets (Figure 1a,b). In chondrogenic conditions, cells aggregated in nodule-like formations (Figure 1c,d) and red-stained calcium deposits were found in osteogenic conditions (Figure 1e,f).

### 3.2. Spheroid Formation

As shown in Figure 2 and Figure 3, after 7 days of culture, the size of the spheroids was stabilized and 1 spheroid per well was formed. The sizes of the final spheroids ranged between 392 and 427 μm.

### 3.3. Neurogenic Differentiation

#### 3.3.1. Ovine Mesenchymal Stem Cell 2D Cultures

The differentiation of oBM-MSCs into neuron-like cells was assessed by Nissl bodies staining in cells subjected to neurogenic differentiation, as well as by analyzing the expression of three neuronal markers: *NEFM*, *MAP2,* and *TUBB3*.

The presence of Nissl bodies in the soma of neuron-like cells was confirmed using Cresyl Violet staining (Figure 4). Regarding the expression of neuronal markers, *TUBB3* and *MAP2* displayed higher expression levels in neuron-like cells in comparison to the cells in growth conditions. *NEFM*, conversely, showed lower expression levels in cells subjected to neurogenic differentiation (Table 1). Overall, the presence of Nissl bodies and the increased expression of two neuronal markers confirmed the neurogenic differentiation of oBM-MSCs.

#### 3.3.2. Spheroids

The evaluation of neurogenic differentiation in spheroids was carried out in the same way as in oBM-MSCs: Cresyl violet staining was performed along with the expression analysis of *NEFM*, *TUBB3,* and *MAP2* neuronal markers.

Together with the presence of Nissl bodies, spheroids under neurogenic differentiation conditions displayed a kind of prolongation on its contour that could be associated with neuronal morphology (Figure 5). Moreover, in neuron-like differentiated spheroids, the expression of the three neuronal markers was increased, albeit slightly, compared to the spheroids in basal conditions (Table 2). Therefore, the neurogenic differentiation of spheroids was also confirmed.

### 3.4. Levels of PrP^Sc^ after Prion Infection

#### 3.4.1. Ovine Mesenchymal Stem Cells and Neuron-like Cells Infected with Scrapie

The PrP^Sc^ signal was measured after the infection with scrapie brain homogenate in oBM-MSCs and neuron-like cells at 2, 5, and 8 days post-inoculation. As shown in Figure 6, the levels of PrP^Sc^ in oBM-MSCs cultured in 2D decreased over time, whereas in neuron-like cells, the levels were stably maintained throughout the study period, with a slight decrease at 8 dpi.

#### 3.4.2. Spheroids in Growth and Neurogenic Conditions Infected with Scrapie

The levels of PrP^Sc^ detected by ELISA were also analyzed in scrapie-infected spheroids in growth and neurogenic conditions. Unlike 2D oBM-MSCs, spheroids in growth conditions showed stable levels of PrP^Sc^ over time (Figure 7a). In spheroids cultured in neurogenic conditions, a slightly decrease in the PrP^Sc^ signal was found at 5 dpi, but later, t 8 dpi, PrP^Sc^ levels were recovered a (Figure 7b). The presence of PrP^Sc^ in scrapie-infected spheroids was also confirmed by immunocytochemistry (Figure 8).

### 3.5. Viability after Prion Infection

#### 3.5.1. oBM-MSC and Neuron-like Cell 2D Cultures

The effect of prion infection on cell viability was assessed at three infection times. oBM-MSCs infected with scrapie-positive inoculum showed increased viability compared to the cells infected with negative inoculum and the non-inoculated controls at 2 dpi, whereas at 5 dpi, a significant decrease in viability was observed in scrapie- and negative-infected cells compared to the controls. At 8 dpi, the levels of viability in the cells infected with positive and negative inocula were recovered, and were similar to the non-inoculated controls (Figure 9a).

In neuron-like cells, an initial increase in cell viability was also observed in scrapie-infected cells along with a significant decrease at 5 dpi. Conversely to oBM-MSCs, this decrease was maintained at 8 dpi (Figure 9b).

#### 3.5.2. Spheroids in Growth and Neurogenic Conditions

Cell viability was also assessed in spheroids in both growth and neurogenic differentiation conditions.

In growth conditions, spheroids infected with the negative inoculum displayed increasing cell viability over time. The ones infected with positive inoculum, however, showed an increased cell viability at 5 dpi, but afterwards, at 8 dpi, the viability decreased greatly (Figure 10a).

In neurogenic conditions, spheroids infected with the negative inoculum also showed high levels of cell viability at the three infection times. Contrary to what was observed in spheroids in growth conditions, spheroids infected with scrapie-positive inocula displayed increments in cell viability over time (Figure 10b).

## 4. Discussion

Three-dimensional in vitro culture systems have arisen as a novel approach for cell cultures. These systems are able to more effectively mirror the microenvironment and interactions of the cells in vivo, making them useful tools for creating in vitro disease models that reliably reproduce disease-inherent characteristics. Our group previously evaluated the effect of scrapie prion infection in oBM-MSCs cultured in 2D in basal and neurogenic conditions [35]. In order to determine whether the three-dimensional conditions induce changes in the way in which these cells react to prion infection, in the present study, we assessed the effect of scrapie prion infection in oBM-MSCs and oBM-MSC-derived neuron-like cells, cultured in two-dimensional monolayer conditions and as spheroids, analyzing the ability to maintain or propagate PrP^Sc^ and possible prion toxicity that could affect cell viability.

Mesenchymal stem cells were the first to be characterized. Their plastic adhesion ability and the tri-lineage differentiation confirmed the nature of these cells [37,39]. The ability to differentiate in vitro into neuron-like cells was also confirmed by morphological analysis in 2D cultures and Nissl staining in both 2D cultures and spheroids. However, the expression of neuronal markers was different between 2D and 3D conditions. Although we cannot discard a potential effect of the RNA isolation method on these differences, in both 2D and 3D cultures, basal and differentiating conditions were treated with the same methodology and the amplification conditions did not allow the amplification of genomic DNA that could possibly cause contamination. It has been reported that undifferentiated human MSCs from different sources, including bone marrow, are able to express different neuronal markers including *TUBB3*, *NEFM,* and *MAP2,* which would explain the slight increase in the expression of some neurogenic markers between basal and neurogenic conditions. This fact implies that the expression analysis of neuronal markers should be complemented with other techniques to evaluate the neural differentiation of MSCs, such as morphological changes and specific stains [45,46], as we already did. The overexpression of *MAP2* was much lower in differentiated spheroids than in 2D cultures; variations in the expression levels of *MAP2* and *TUBB3* between 2D and 3D cultures have also been described in the neural induction of neural progenitor cells obtained from human iPSC [47].

Murine bone marrow-derived MSCs can be infected and propagate the Fukuoka-1 human prion strain [48,49], and a persistent propagation of the mouse-adapted variant CJD and Fukuoka-1 strains has been described in spleen-derived murine stromal cells [50]. In accordance with our previous study [35], we observed a decline in PrP^Sc^ levels in 2D cultures of oBM-MSCs infected with scrapie throughout the duration of the culture, whereas neuron-like cells displayed steady PrP^Sc^ levels over time. These results confirm that oBM-MSCs cultured in two-dimensional conditions are less permissive to scrapie infection compared to neuron-like cells, which seem to be able to uptake and maintain scrapie infection over time.

Regarding 3D in vitro models for prion diseases, only human cerebral organoids have been described to uptake and propagate sCJD prions [33]. Within MSC-derived spheroids, a specific microenvironment more similar to in vivo conditions is formed in which the diffusion inwards of nutrients and gases, and outwards of metabolic wastes, is limited [51,52]. Moreover, a decreased cell size, cell cycle quiescence and reduced energy metabolism have been found in MSC spheroids compared with MSCs in monolayer conditions [51,53,54]. The differences in the microenvironment of MSCs in 2D and 3D conditions could explain the distinct results observed between oBM-MSCs infected with scrapie on a culture plate and scrapie-infected spheroids in growth conditions. Contrary to 2D cultures, spheroids in growth conditions were able to maintain stable levels of PrP^Sc^ over time. On the other hand, like neuron-like cells, spheroids in neurogenic conditions also took up and propagated PrP^Sc^ levels. These results indicate that the 3D microenvironment makes oBM-MSCs more permissive to prion infection when cultured in growth conditions, and maintains the ability of neuron-like cells to absorb and propagate scrapie prions.

Although primary neuronal cultures show toxicity after prion propagation in vitro that varies in a strain- and neuronal type-specific manner [55,56], murine MSCs are able to propagate prions in growth conditions without signs of toxicity [48,49,50]. In our previous study, oBM-MSCs and their neuron-like cell derivatives, infected with brain homogenates of healthy and scrapie-infected sheep, displayed increasing levels of cell viability throughout the duration of the culture in comparison to non-inoculated cells, suggesting that brain inocula may contain factors that stimulate oBM-MSC proliferation [35]. Similarly, in the present study, scrapie-infected oBM-MSCs showed higher cell viability levels than the non-inoculated controls just after removing the inocula (2 dpi); however, we did not observe this increase in cells inoculated with negative inoculum. It seems that MSC division is more stimulated with scrapie brains. In fact, it has been described that MSCs can migrate to brain lesions caused by prions, and this migration is mediated by different chemoattractive factors [57,58,59]. Similarly, neuron-like cells infected with scrapie displayed an increase in cell viability only after being in contact with the inocula, confirming our previous data. However, in this study, toxicity was observed afterwards. Therefore, the effect of brain inocula in oBM-MSC proliferation seems to vary between cultures. This variation could be explained by the cellular heterogeneity found in MSCs, in which donor, tissue source, culture environment, isolation methods, and passage can affect the phenotype [60,61], making distinct cultures react slightly different to CNS homogenates. However, we have to keep in mind that inocula of both negative and positive brains are also different. The amounts of initial PrP^Sc^ could be different, which would affect the reaction of MSC to infection. Nevertheless, the early response to scrapie infection was the same in both studies, an increment of cell viability, which means a higher proliferation potential either in growth or neurogenic conditions when cultured on plates after being in contact with scrapie tissues. The decrease in cell viability observed afterwards could be a consequence of prion toxicity.

Ovine MSC spheroids inoculated with brain homogenates of healthy and scrapie-infected sheep displayed different viability patterns than the monolayer-cultured cells. The negative inoculum increased the viability of spheroids in both growth and neurogenic conditions; however, this increase was not observed in spheroids infected with scrapie, which remained at viability levels similar to the controls. Compared to monolayer-cultured MSCs, human MSC spheroids show higher cell survival and enhanced cell yield and stemness [62,63,64]. The enhanced cell viability of spheroids seems to be mediated by the induction of autophagy and the suppression of reactive oxygen species (ROS) [64]. Remarkably, an impairment of the autophagy process has been described in scrapie disease [65,66]. The decreased viability observed in scrapie-infected spheroids in growth conditions, with respect to those inoculated with negative brain material could be due to a dysfunction of the autophagy mechanism caused by prion infection that could counteract the positive effect exerted by neurotrophic factors in the brain. These results suggest that in most cases, 3D conditions improve the viability of inoculated oBM-MSCs, as infected spheroids at least maintain the viability observed in controls, but prions exert their toxicity by limiting the growth potential of spheroids stimulated with neurotrophic factors.

## 5. Conclusions

In conclusion, oBM-MSC-derived spheroids in growth and neurogenic conditions seem to be able to uptake and propagate scrapie prions and mimic prion toxicity, making this three-dimensional approach a potential in vitro model to study prion disease mechanisms and therapeutics in a more in vivo-like environment. Nevertheless, further studies using a larger number of cultures are still necessary to confirm the reproducibility of this in vitro model, along with the analysis of the PrP^Sc^ profile generated in both oBM-MSCs and spheroids in growth and neurogenic conditions.

## Figures and Tables

**Figure 1 animals-13-01043-f001:**
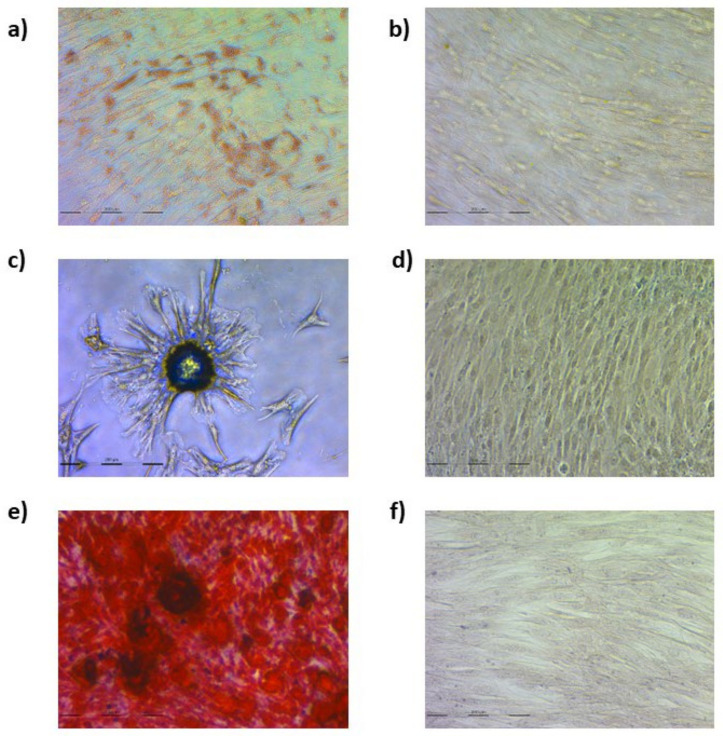
oBM-MSC staining for adipogenic, chondrogenic, and osteogenic differentiation. Oil red O staining of cells cultured for 8 days in adipogenic differentiation medium (**a**) and basal medium (**b**); alcian blue staining of cells cultured for 15 days in chondrogenic (**c**) and basal medium (**d**); and alizarin red staining of cells cultured for 21 days in osteogenic differentiation medium (**e**) and basal medium (**f**).

**Figure 2 animals-13-01043-f002:**
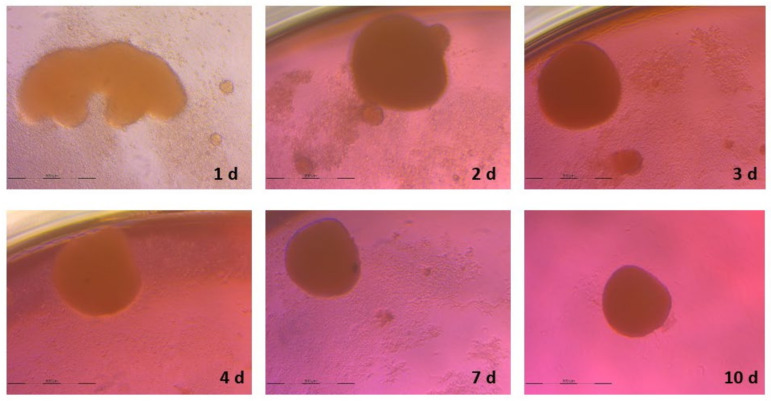
Timeline images of spheroid formation from 1 day to 10 days of culture in basal conditions.

**Figure 3 animals-13-01043-f003:**
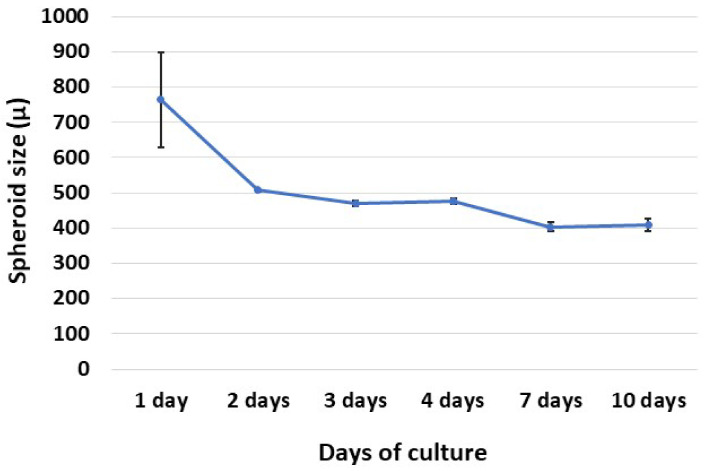
Line chart showing the size of the spheroids (µ) throughout 10 days of culture in basal conditions. The data are presented as mean values ± SEM (n = 4).

**Figure 4 animals-13-01043-f004:**
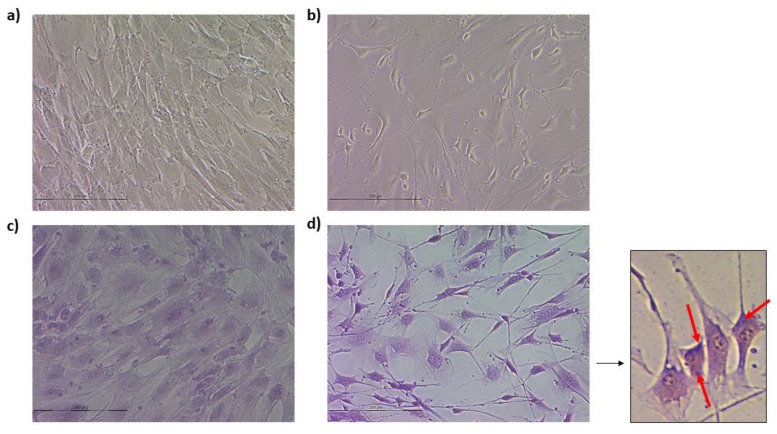
oBM-MSCs in basal (**a**) and neurogenic differentiation (**b**) conditions. Cresyl Violet staining of oBM-MSCs in basal medium (**c**) and oBM-MSC derived neuron-like cells (**d**) 3 days after neurogenic induction. Nissl bodies present in neuron-like cells are marked with red arrows.

**Figure 5 animals-13-01043-f005:**
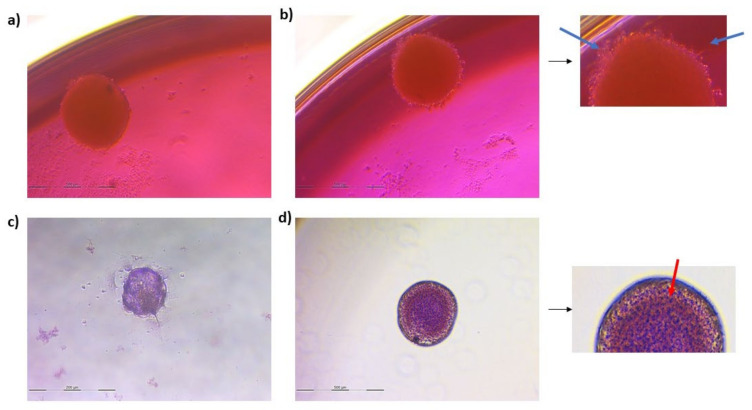
Spheroids in basal (**a**) and neurogenic differentiation (**b**) conditions. Cresyl violet staining of spheroids in basal (**c**) and neurogenic (**d**) conditions 3 days after neuronal induction. Blue arrows indicate the prolongations formed on the contours of the spheroid and the red arrow marks the Nissl bodies.

**Figure 6 animals-13-01043-f006:**
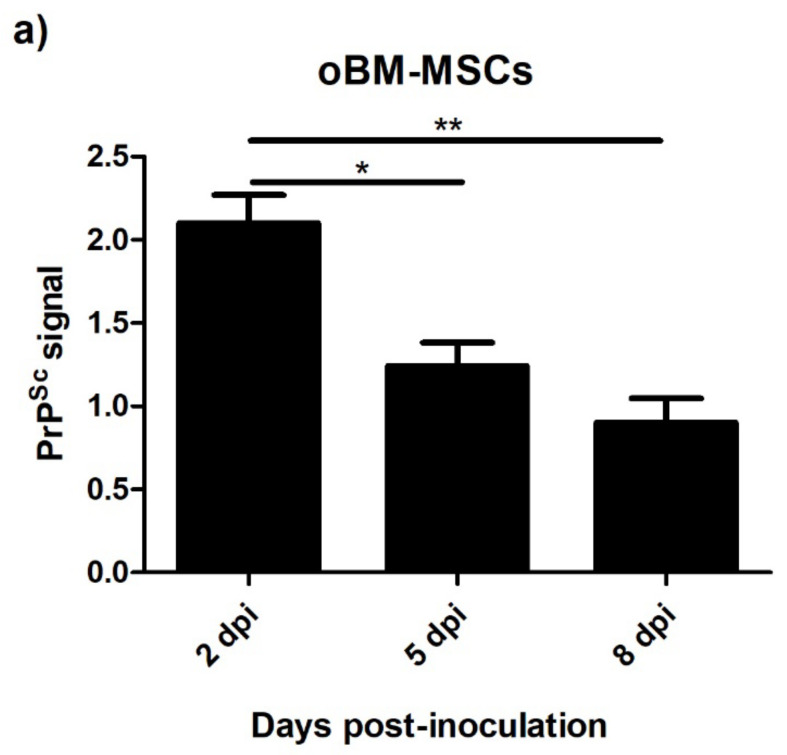
PrP^Sc^ levels measured by ELISA in oBM-MSCs (**a**) and neuron-like cells (**b**) 2, 5, and 8 days after prion infection. Data are shown as mean values ± SEM. * *p* < 0.05; ** *p* < 0.01.

**Figure 7 animals-13-01043-f007:**
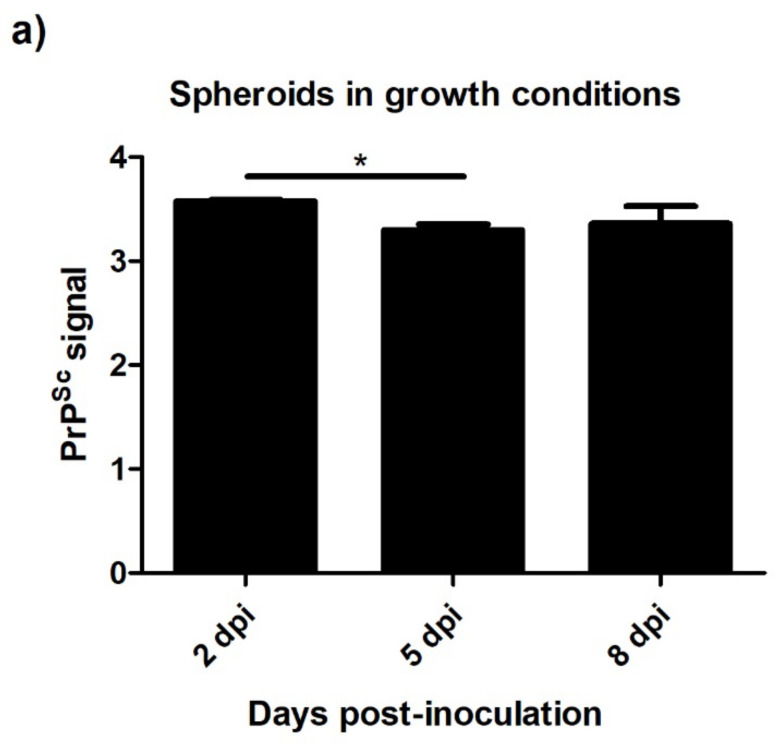
PrP^Sc^ levels detected by ELISA in spheroids in growth (**a**) and neurogenic (**b**) conditions 2, 5, and 8 days after prion infection. Data are shown as mean values ± SEM. * *p* < 0.05; *** *p* < 0.001.

**Figure 8 animals-13-01043-f008:**
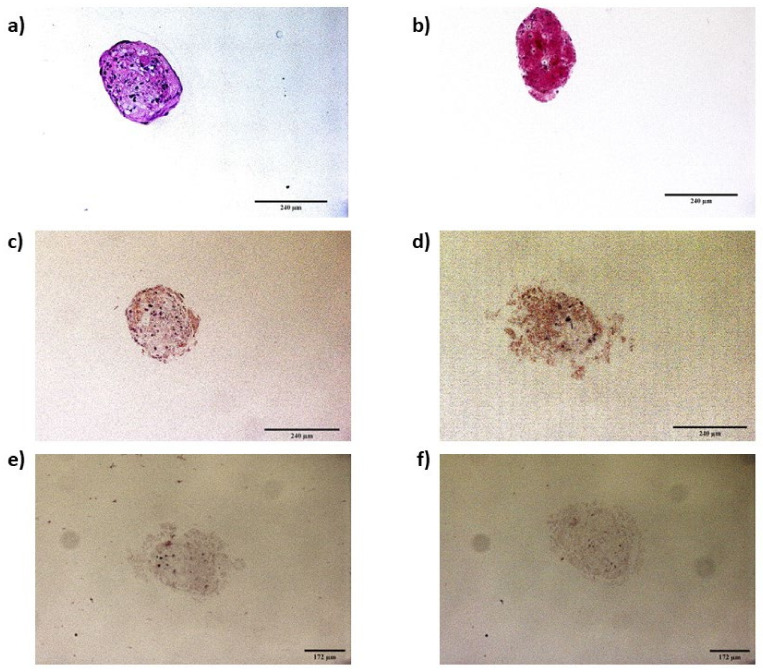
Hematoxylin–eosin staining of spheroids in growth (**a**) and neurogenic (**b**) conditions. PrP^Sc^ immunostaining (brown deposits) of scrapie-infected spheroids in growth (**c**) and neurogenic differentiation (**d**) conditions, and their respective background controls with omission of the primary antibody (**e**,**f**).

**Figure 9 animals-13-01043-f009:**
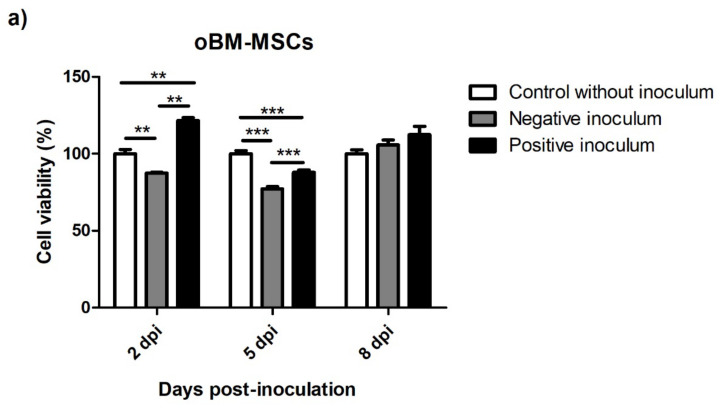
Cell viability of oBM-MSCs (**a**) and neuron-like cells (**b**) 2, 5, and 8 days after prion infection. The viability values are presented as percentages (%), considering the viability of the controls without inoculum as 100% for each infection time. Data are shown as mean values ± SEM. * *p* < 0.05; ** *p* < 0.01; *** *p* < 0.001.

**Figure 10 animals-13-01043-f010:**
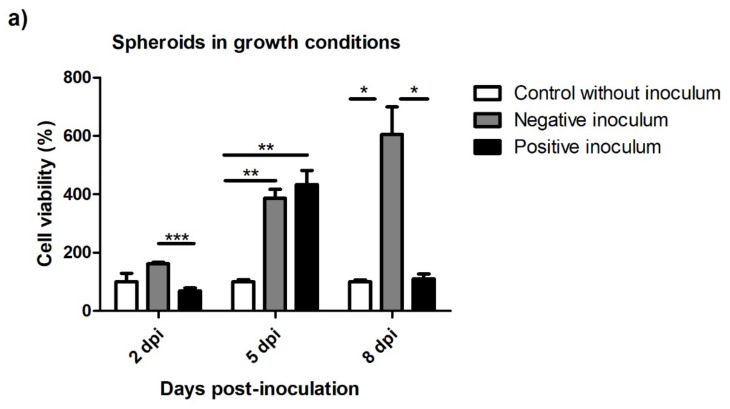
Cell viability of spheroids in growth (**a**) and neurogenic (**b**) conditions 2, 5, and 8 days post-inoculation. The viability values are presented as percentages (%), considering the viability of the controls without inocula as 100% for each infection time. Data are shown as mean values ± SEM. * *p* < 0.05; ** *p* < 0.01; *** *p* < 0.001.

**Table 1 animals-13-01043-t001:** Expression levels of neuronal markers *NEFM*, *MAP2,* and *TUBB3* in oBM-MSCs and neuron-like cells 3 days after neurogenic induction. Relative expression levels are shown in terms of 2^−∆∆Ct^.

Gene	Expression Levels (2 ^−∆∆Ct^) in oBM-MSCs	Expression Levels (2 ^−∆∆Ct^) in Neuron-like Cells
*TUBB3*	1	1.46
*NEFM*	1	0.50
*MAP2*	1	321.63

**Table 2 animals-13-01043-t002:** Expression levels of neuronal markers *NEFM*, *MAP2,* and *TUBB3* in spheroids in basal and neurogenic differentiation conditions 3 days after neurogenic induction. Relative expression levels are shown in terms of 2^−∆∆Ct^.

Gene	Expression Levels (2^−∆∆Ct^) in Spheroids in Basal Conditions	Expression Levels (2^−∆∆Ct^) in Spheroids in Neurogenic Differentiation Conditions
*TUBB3*	1	1.21
*NEFM*	1	1.13
*MAP2*	1	2.24

## Data Availability

The data presented in this study are contained within the article and supplementary materials.

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
