# Peer review of "Susceptibility of Ovine Bone Marrow-Derived Mesenchymal Stem Cell Spheroids to Scrapie Prion Infection"

_animals, 2023, doi:10.3390/ani13061043_

Round 1
Reviewer 1 Report
The manuscript by Adelaida Hernaiz, Paula Cobeta, Belén Marín, Francisco J. Vázquez, Juan J. Badiola, Pilar Zaragoza, Rosa Bolea and Inmaculada Martín-Burriel, entitled “Susceptibility of ovine bone marrow-derived mesenchymal stem cell spheroids to scrapie prion infection”, describes the study on three-dimensional spheroids of ovine mesenchymal stem cells cultured in different conditions as a potential model for prion and prion diseases studies.
The manuscript is understandable. I didn’t find any mistakes, however I am not the English native speaker.
There are some editorial mistakes in the text, like extra spaces in the discussion section.
Introduction is a good background for the study.
Methods.
The methods allow the reader to reconstruct the research. But I have some questions.
It is not sufficiently clear what the size of the groups was in each part of the experiment. It would have been much clearer if the authors had prepared a table (in methods or as supplementary materials ) with detailed information on the numbers and technical replicates in each part of the experiment. Please add the a table to clarify the group and repetition sizes for the entire experiment (every part of it).
How many separate samples were used in the gene expression experiment? It is not clear how large the groups being compared were.
Did the authors perform a gene expression experiment in triplicates?
Did the Authors checked the RNA quality and quantity?
How did the Authors isolated RNA from spheroids? From the whole spheroids?
The gene expression experiment was prepared differently for 2D neuron-like cell cultures and spheroids. I shouldn’t be so. Gene expression results are influenced by many factors and the method of isolation is one of them. Did the authors check the level of RNA contamination by DNA (noRT experiment)? Did the primers amplify cDNA-only products (were specific only to cDNA)?
The Authors used means and the Student’s t-test and calculated the statistical significance. The p level was used for ELISA experimental groups and cell viability assay. But it looks like the compered groups were very small (it is not clear enough) and it is not clear whether they were equally numerous. Using statistical tests with too small groups can lead to too hasty conclusions.
In gene expression experiments and ELISA, did the authors check the differences between the cells in the outer layers of the spheroid and in the middle of the spheroid or treated the spheroid as a whole?
How many spheroids were stained in scrapie-infected group and background controls?
Results and discussion
The results are well presented and properly described.
All pictures are of good quality and make an interesting addition to the results. Graphs are easy to read.
With study groups small in size and different methods of isolation, it is debatable to compare the results between such groups. The use of endogenous controls does not completely eliminate such differences. Especially if the purity of the RNA was not checked, nor if the primers could amplify with DNA as well with cDNA. Please clarify that or comment that in the discussion.
Did the Authors observed differences in PrPSc immunostaining between single, different scrapie-infected spheroids?
Conclusions
The conclusion are supported by the results. However, the Authors concluded, that oBM-MSC-derived spheroids can be a potential in vitro model to study prion disease mechanisms. Creating a model is a long-term process. It requires repeating the entire experiment at least several times and checking the reproducibility of all the results. Especially since cell cultures are rarely perfectly reproducible because it is very difficult to ensure identical conditions.
How many times did the Authors repeated the whole experiment from the beginning? If this was just a single experiment, these are more of a preliminary results. And this should be emphasized - also in the manuscript title.
The references
Among 63 references, 33 were published before 2017 and the rest have been published in the last 5 years. All cited references are relevant to the research. I didn’t detect any inappropriate self-citations.
In general, the research was quite properly conducted and manuscript is well written. However, the experiment looks like preliminary studies. The authors need to clarify the methodological issues raised. The manuscript can be reconsider for publication after revision.
Author Response
We would like to thank this reviewer for the comments and suggestions made. The answers to the reviewer’s comments are written in italics.
Methods.
The methods allow the reader to reconstruct the research. But I have some questions.
It is not sufficiently clear what the size of the groups was in each part of the experiment. It would have been much clearer if the authors had prepared a table (in methods or as supplementary materials) with detailed information on the numbers and technical replicates in each part of the experiment. Please add the table to clarify the group and repetition sizes for the entire experiment (every part of it).
Thank you for the comment. A supplementary table (supplementary Table S1) has been prepared including the number of technical replicates used in each experiment. In the materials and methods section, subsection 2.1, a sentence has been added citing the table to clarify the experiments conducted (lines 130-132).
In addition, in the materials and methods section, subsection 2.4.3 (lines 184-187), it has also been clarified and detailed the number of technical replicates used in the expression analysis of neuronal markers as we did not include it previously when writing this section.
How many separate samples were used in the gene expression experiment? It is not clear how large the groups being compared were.
For oBM-MSCs and spheroids, 2 technical replicates were used for basal and neurogenic conditions. For 2D cultures, each replicate consisted of 2 wells of a P-6 well culture plate. On the other hand, each replicate for expression analysis of spheroids consisted of 42 single spheroids treated as a pool. This information has been added in supplementary Table S1 and subsection 2.4.3 of the materials and methods section.
Did the authors perform a gene expression experiment in triplicates?
Yes, each sample was analysed per triplicate. This information has been added in supplementary Table S1.
Did the Authors checked the RNA quality and quantity?
In 2D cultures of oBM-MSCs and neuron-like cells the RNA quality and quantity were not checked as the Cells-to-cDNATM II kit was used. This kit is specific for mammal cells and allows the obtention of cDNA directly from cells without requiring a previous RNA purification and warranting the degradation of genomic DNA.
Regarding the spheroids, RNA quality and quantity were checked with Nanodrop and Qubit instruments to ensure the obtention of optimal RNA. In the materials and methods section, subsection 2.4.3, a sentence has been added explaining this information (lines 187-190).
How did the Authors isolated RNA from spheroids? From the whole spheroids?
We isolated total RNA from a pool of 42 spheroids in each replicate. The number of cells included in a single spheroid is similar to the one cultured on a well of a 6-well plate, however the amount of RNA obtained was much lower, probably due to the reduced activity and metabolism of MSCs in spheroids compared to monolayer cultured ones (Jauković, A. et al. 2020; Liu, Y. et al. 2017; Zhang, Q. et al. 2012; Tae, J.Y. et al. 2021).
- Jauković, A.; Abadjieva, D.; Trivanović, D.; Stoyanova, E.; Kostadinova, M.; Pashova, S.; Kestendjieva, S.; Kukolj, T.; Jeseta, M.; Kistanova, E.; et al. Specificity of 3D MSC Spheroids Microenvironment: Impact on MSC Behavior and Properties. Stem cell Rev. reports 2020, 16, 853–875, doi:10.1007/S12015-020-10006-9.
- Liu, Y.; Muñoz, N.; Tsai, A.C.; Logan, T.M.; Ma, T. Metabolic Reconfiguration Supports Reacquisition of Primitive Phenotype in Human Mesenchymal Stem Cell Aggregates. Stem Cells 2017, 35, 398–410, doi:10.1002/STEM.2510.
- Zhang, Q.; Nguyen, A.L.; Shi, S.; Hill, C.; Wilder-Smith, P.; Krasieva, T.B.; Le, A.D. Three-Dimensional Spheroid Culture of Human Gingiva-Derived Mesenchymal Stem Cells Enhances Mitigation of Chemotherapy-Induced Oral Mucositis. Stem Cells Dev. 2012, 21, 937–947, doi:10.1089/SCD.2011.0252.
- Tae, J.Y.; Lee, H.; Lee, H.; Song, Y.; Park, J.-B. Morphological Stability, Cellular Viability and Stem Cell Marker Expression of Three-Dimensional Cultures of Stem Cells from Bone Marrow and Periodontium. Biomed. reports 2021, 14, 9, doi:10.3892/BR.2020.1385.
The gene expression experiment was prepared differently for 2D neuron-like cell cultures and spheroids. I shouldn’t be so. Gene expression results are influenced by many factors and the method of isolation is one of them. Did the authors check the level of RNA contamination by DNA (noRT experiment)? Did the primers amplify cDNA-only products (were specific only to cDNA)?
Thank you for the comment. We are aware that the isolation method used was different between 2D and 3D conditions and as the reviewer points it could influence our results. This limitation has been explained in the discussion section lines 431-451. Different methods were chosen because the kit selected for each condition was the one that allowed us to obtain the best yield in the RT-qPCR experiments. On the one hand, the Cells-to-cDNA kit allows RNA retrieval with enough quality for RT-qPCR from a limited number of cells, on the other hand, Trizol seems to be necessary for RNA retrieval from spheroids, as many articles that perform RNA analysis in spheroids use it (Barbone, D. et al. 2016; Mihara, H. et al. 2019; Muñoz-Galindo, L. et al. 2019). We were not able to obtain enough RNA from single spheroids using the kit as it was difficult to disaggregate, and that is why, following the bibliography, we performed RNA extraction from 42 pooled spheroids treated with Trizol (Muñoz-Galindo, L. et al. 2019).
The RNA contamination by DNA was not specifically checked as both kits used ensure the proper degradation of genomic DNA by the use of DNase. Nevertheless, the primers have been used previously in other studies from our group (reference 37) and are cDNA specific. Primers were designed in different exons, which allows to differentiate between the amplification of cDNA and genomic DNA. After RT-qPCR amplification, a dissociation curve was performed and every sample showed a single amplicon peak.
- Barbone, D.; Van Dam, L.; Follo, C.; Jithesh, P. V.; Zhang, S.D.; Richards, W.G.; Bueno, R.; Fennell, D.A.; Broaddus, V.C. Analysis of Gene Expression in 3D Spheroids Highlights a Survival Role for ASS1 in Mesothelioma. PLoS One 2016, 11, doi:10.1371/journal.pone.0150044.
- Mihara, H.; Kugawa, M.; Sayo, K.; Tao, F.; Shinohara, M.; Nishikawa, M.; Sakai, Y.; Akama, T.; Kojima, N. Improved Oxygen Supply to Multicellular Spheroids Using A Gas-Permeable Plate and Embedded Hydrogel Beads. Cells 2019, 8, doi:10.3390/CELLS8060525.
- Muñoz-Galindo, L.; Melendez-Zajgla, J.; Pacheco-Fernández, T.; Rodriguez-Sosa, M.; Mandujano-Tinoco, E.A.; Vazquez-Santillan, K.; Castro-Oropeza, R.; Lizarraga, F.; Sanchez-Lopez, J.M.; Maldonado, V. Changes in the Transcriptome Profile of Breast Cancer Cells Grown as Spheroids. Biochem. Biophys. Res. Commun. 2019, 516, 1258–1264, doi:10.1016/J.BBRC.2019.06.155.
The Authors used means and the Student’s t-test and calculated the statistical significance. The p level was used for ELISA experimental groups and cell viability assay. But it looks like the compered groups were very small (it is not clear enough) and it is not clear whether they were equally numerous. Using statistical tests with too small groups can lead to too hasty conclusions.
The number of technical replicates used in the ELISA and cell viability experiments has been included in the supplementary Table S1. In the ELISA assay, all groups were equally numerous in 2D and 3D conditions, and for the MTT assay, the groups were equally numerous in each condition, with 8 technical replicates per group in 2D conditions and 4 technical replicates per group in 3D conditions.
Although it would be always better to have the maximum number of technical replicates, we think that in the case of MTT and ELISA assays the number is adequate. To reproduce the model, it would be necessary to test the ability of other cultures (MSCs obtained from other sheep) to confirm that the response is similar. This limitation has been explained in the conclusions section lines 522-526.
In gene expression experiments and ELISA, did the authors check the differences between the cells in the outer layers of the spheroid and in the middle of the spheroid or treated the spheroid as a whole?
Spheroids are two small to separate inner and outer layers, as we said before it has been necessary to pool 42 spheroids for RNA purification. Although ELISA was performed with single spheroids, we don’t have the technology to micromanipulate the spheroids. Only immunohistochemistry allowed us to detect PrPSc in all layers.
How many spheroids were stained in scrapie-infected group and background controls?
4 scrapie-infected spheroids in basal conditions, 4 scrapie-infected spheroids in differentiation conditions and 4 background controls. This information has been included in supplementary Table S1.
Results and discussion
The results are well presented and properly described.
All pictures are of good quality and make an interesting addition to the results. Graphs are easy to read.
With study groups small in size and different methods of isolation, it is debatable to compare the results between such groups. The use of endogenous controls does not completely eliminate such differences. Especially if the purity of the RNA was not checked, nor if the primers could amplify with DNA as well with cDNA. Please clarify that or comment that in the discussion.
As previously mentioned, the RNA purity was checked in spheroids and the primers used were cDNA specific. Regarding the comparison of the results of the gene expression analysis between 2D and 3D conditions, the limitation of the different method of isolation has been clarified in the discussion section lines 431-451. In addition, we have also discussed the expression results of our work with others (lines 431-451) (Wu, S.H. et al. 2021; Foudah, D. et al. 2014; Chandrasekaran, A. et al. 2017).
- Wu, S.H.; Liao, Y.T.; Huang, C.H.; Chen, Y.C.; Chiang, E.R.; Wang, J.P. Comparison of the Confluence-Initiated Neurogenic Differentiation Tendency of Adipose-Derived and Bone Marrow-Derived Mesenchymal Stem Cells. Biomedicines 2021, 9, doi:10.3390/BIOMEDICINES9111503.
- Foudah, D.; Monfrini, M.; Donzelli, E.; Niada, S.; Brini, A.T.; Orciani, M.; Tredici, G.; Miloso, M. Expression of Neural Markers by Undifferentiated Mesenchymal-like Stem Cells from Different Sources. J. Immunol. Res. 2014, 2014, doi:10.1155/2014/987678.
- Chandrasekaran, A.; Avci, H.X.; Ochalek, A.; Rösingh, L.N.; Molnár, K.; László, L.; Bellák, T.; Téglási, A.; Pesti, K.; Mike, A.; et al. Comparison of 2D and 3D Neural Induction Methods for the Generation of Neural Progenitor Cells from Human Induced Pluripotent Stem Cells. Stem Cell Res. 2017, 25, 139–151, doi:10.1016/J.SCR.2017.10.010.
Did the Authors observed differences in PrPSc immunostaining between single, different scrapie-infected spheroids?
Overall, all the stained spheroids showed similar immunostaining, although it would be interesting to perform this technique in a larger sample size and evaluate quantitatively the PrPSc staining.
Conclusions
The conclusions are supported by the results. However, the Authors concluded, that oBM-MSC-derived spheroids can be a potential in vitro model to study prion disease mechanisms. Creating a model is a long-term process. It requires repeating the entire experiment at least several times and checking the reproducibility of all the results. Especially since cell cultures are rarely perfectly reproducible because it is very difficult to ensure identical conditions.
How many times did the Authors repeated the whole experiment from the beginning? If this was just a single experiment, these are more of a preliminary results. And this should be emphasized - also in the manuscript title.
Regarding the question of if this was a single experiment, before obtaining these results, we performed several assays to ensure the efficiency and suitability of each technique used in this study for the characterization of spheroids and for the evaluation of their prion infection potential and cell viability after scrapie infection.
Nevertheless, we agree with the reviewer that further studies are necessary mainly using cells obtained from different donors. This fact has been pointed in the conclusions section lines 522-526.
The references
Among 63 references, 33 were published before 2017 and the rest have been published in the last 5 years. All cited references are relevant to the research. I didn’t detect any inappropriate self-citations.
In general, the research was quite properly conducted and manuscript is well written. However, the experiment looks like preliminary studies. The authors need to clarify the methodological issues raised. The manuscript can be reconsider for publication after revision.
We thank the reviewer for all the comments. We hope to have addressed all the methodological issues kindly marked by this reviewer.
Reviewer 2 Report
The paper of Hernaiz et al., describes a methodology for culture of ovine bone-marrow mesenchymal stem cell spheroids and shows their capacity to propagate scrapie prion. Several classical cell culture systems exist to propagate different prion strains from different species. However, organoid and spheroid models that are for certain degree closer to tissues organization in vivo, susceptible to prion infection are very rare. The present work has the merit to develop this kind of model for infection with scrapie prion. The paper shows that in neurogenic conditions both monolayer cells and spheroids can propagate the scrapie prion used. However, in growth conditions prion infection decreased over time in monolayer oBM-MSCs, whereas spheroids are able to maintain PrPSc propagation. Overall, the paper is interesting and well presented.
Below are listed some questions, suggestions and clarifications which need to be brought :
1- Line 21-22: Authors say that 3-D approach appears… in a microenvironment most similar to in vivo conditions. Authors should attenuate their claim as even 3D culture could never be similar to in vivo situation. It could only approach better than 2D cell culture the in vivo situation. Then an appropriate word like approaching in vivo conditions might be more adequate.
2- Lines 168-177: Why RNA extraction protocols are different between 2D neuron-like cells culture and spheroids?
3- Usually to reveal PrPSc, samples are treated with Guanidium hydrochloride and Proteinase K to unmask PrPSc epiope and to eliminate PrPC for background respectively. The protocol described in the present work did not contain these steps and I wonder how one can detect PrPsc without these treatments.
4- The increase of RNA levels of neurogenic markers seems not very strong between basal conditions and neurogenic one’s. Also, the level of NEFM even decreased between oBM-MSCs cells and their counterpart in neuron-like cells. Please clarify.
5- Lines 424-426: Authors should make attention to the vocabulary used in sentences such as “took up and maintained PrPSc levels” or “to absorb and retain scrapie prions”. Indeed, spheroids in neurogenic conditions took up or absorb PrPSc or scrapie prions, but they do not only maintained or retain them, but they amplify and propagate them. The term prion propagation is much more adequate in this context.
6- Overall, PrPSc level measurements made with IDEXX-Herdcheck Elisa test are suitable for this study, analysis of the PrPres profile by western blotting could be of interest for a complete analysis to check whether the PrPres biochemical profile of the scrapie prion generated in neuron-like cells and in spheroids in neurogenic conditions is similar to the original inoculum.
Author Response
We would like to thank this reviewer for the kind comments and suggestions made. The answers to the reviewer’s comments are written in italics.
Below are listed some questions, suggestions and clarifications which need to be brought:
- Line 21-22: Authors say that 3-D approach appears… in a microenvironment most similar to in vivo conditions. Authors should attenuate their claim as even 3D culture could never be similar to in vivo situation. It could only approach better than 2D cell culture the in vivo situation. Then an appropriate word like approaching in vivo conditions might be more adequate.
As the reviewer suggested, we have changed the term “more similar to” for “approaching” in lines 21-22.
- Lines 168-177: Why RNA extraction protocols are different between 2D neuron-like cells culture and spheroids?
Thank you for the comment. As the reviewer pointed, the RNA isolation method used was different between 2D and 3D conditions. Different methods were chosen because the kit selected for each condition was the one that allowed us to obtain the best yield in the RT-qPCR experiments. On the one hand, the Cells-to-cDNA kit allows RNA retrieval with enough quality for RT-qPCR from a limited number of cells, on the other hand, Trizol seems to be necessary for RNA retrieval from spheroids, as many articles that perform RNA analysis in spheroids use it (Barbone, D. et al. 2016; Mihara, H. et al. 2019; Muñoz-Galindo, L. et al. 2019). We were not able to obtain enough RNA from single spheroids using the Cells to cDNA kit as it was difficult to disaggregate, and that is why, following the bibliography, we performed RNA extraction from 42 pooled spheroids treated with Trizol (Muñoz-Galindo, L. et al. 2019). We are aware that this fact could influence to some extent our gene expression results, and hence, this limitation has been explained in the discussion section lines 431-451.
- Barbone, D.; Van Dam, L.; Follo, C.; Jithesh, P. V.; Zhang, S.D.; Richards, W.G.; Bueno, R.; Fennell, D.A.; Broaddus, V.C. Analysis of Gene Expression in 3D Spheroids Highlights a Survival Role for ASS1 in Mesothelioma. PLoS One 2016, 11, doi:10.1371/journal.pone.0150044.
- Mihara, H.; Kugawa, M.; Sayo, K.; Tao, F.; Shinohara, M.; Nishikawa, M.; Sakai, Y.; Akama, T.; Kojima, N. Improved Oxygen Supply to Multicellular Spheroids Using A Gas-Permeable Plate and Embedded Hydrogel Beads. Cells 2019, 8, doi:10.3390/CELLS8060525.
- Muñoz-Galindo, L.; Melendez-Zajgla, J.; Pacheco-Fernández, T.; Rodriguez-Sosa, M.; Mandujano-Tinoco, E.A.; Vazquez-Santillan, K.; Castro-Oropeza, R.; Lizarraga, F.; Sanchez-Lopez, J.M.; Maldonado, V. Changes in the Transcriptome Profile of Breast Cancer Cells Grown as Spheroids. Biochem. Biophys. Res. Commun. 2019, 516, 1258–1264, doi:10.1016/J.BBRC.2019.06.155.
- Usually to reveal PrPSc, samples are treated with Guanidium hydrochloride and Proteinase K to unmask PrPSc epiope and to eliminate PrPC for background respectively. The protocol described in the present work did not contain these steps and I wonder how one can detect PrPsc without these treatments.
We thank the reviewer for this important comment. When writing the methodology, we forgot to include the proteinase K treatment. All spheroids sections were digested with 4 µg/ml of proteinase K for 15 min at 37 °C. This sentence has been added to the materials and methods section, subsection 2.6, line 254.
- The increase of RNA levels of neurogenic markers seems not very strong between basal conditions and neurogenic one’s. Also, the level of NEFM even decreased between oBM-MSCs cells and their counterpart in neuron-like cells. Please clarify.
It has been reported that undifferentiated human MSCs from different sources, including bone marrow, are able to express different neuronal markers including TUBB3, NEFM and MAP2. This fact implies that the expression analysis of neuronal markers has to be complemented with other techniques to evaluate the neural differentiation of MSCs, such as morphological changes and specific stains (Wu, S.H. et al. 2021 and Foudah, D. et al. 2014). Therefore, the slightly increase of neurogenic markers observed between basal and neurogenic conditions could be due to the simultaneous expression of these markers in both conditions. In addition, variation in expression levels of MAP2 and TUBB3 between 2D and 3D cultures have also been described in neural induction of neural progenitor cells obtained from human iPSC (Chandrasekaran, A. et al. 2017). This explanation has been added to the discussion section lines 431-451.
- Wu, S.H.; Liao, Y.T.; Huang, C.H.; Chen, Y.C.; Chiang, E.R.; Wang, J.P. Comparison of the Confluence-Initiated Neurogenic Differentiation Tendency of Adipose-Derived and Bone Marrow-Derived Mesenchymal Stem Cells. Biomedicines 2021, 9, doi:10.3390/BIOMEDICINES9111503.
- Foudah, D.; Monfrini, M.; Donzelli, E.; Niada, S.; Brini, A.T.; Orciani, M.; Tredici, G.; Miloso, M. Expression of Neural Markers by Undifferentiated Mesenchymal-like Stem Cells from Different Sources. J. Immunol. Res. 2014, 2014, doi:10.1155/2014/987678.
- Chandrasekaran, A.; Avci, H.X.; Ochalek, A.; Rösingh, L.N.; Molnár, K.; László, L.; Bellák, T.; Téglási, A.; Pesti, K.; Mike, A.; et al. Comparison of 2D and 3D Neural Induction Methods for the Generation of Neural Progenitor Cells from Human Induced Pluripotent Stem Cells. Stem Cell Res. 2017, 25, 139–151, doi:10.1016/J.SCR.2017.10.010.
- Lines 424-426: Authors should make attention to the vocabulary used in sentences such as “took up and maintained PrPSc levels” or “to absorb and retain scrapie prions”. Indeed, spheroids in neurogenic conditions took up or absorb PrPSc or scrapie prions, but they do not only maintained or retain them, but they amplify and propagate them. The term prion propagation is much more adequate in this context.
As the reviewer suggested, the terms “maintain” and “retain” have been changed for “propagation” in lines 472-475 of the discussion section.
- Overall, PrPSc level measurements made with IDEXX-Herdcheck Elisa test are suitable for this study, analysis of the PrPres profile by western blotting could be of interest for a complete analysis to check whether the PrPres biochemical profile of the scrapie prion generated in neuron-like cells and in spheroids in neurogenic conditions is similar to the original inoculum.
Thank you for the comment. We would take into account the reviewer’s suggestion for further studies of this in vitro model. Nevertheless, this possible analysis in further studies has been mentioned in the conclusions section, lines 522-526.
Round 2
Reviewer 1 Report
Dear Authors,
Your manuscript entitled “Susceptibility of ovine bone marrow-derived mesenchymal stem cell spheroids to scrapie prion infection” was improved.
Thank you for all your explanations.
All Your modification and supplementary materials has clarified the methodology used and proved that the experiment was well thought out.
While it could be planned to use more technical replicates, the limitations are understandably explained and discussed.
The following comment is meant to be helpful to the Authors:
Comparing the gene expression level of the samples in which RNA isolation and reverse transcription are different may influence the results. Cells-to-cDNA™ II Kit should give reliable results also for spheroids. You should at least try using this Kit also for spheroids. These remark does not discredit the manuscript, but it should help You in Your future studies.
The research was properly conducted and the manuscript is well written. All methodological issues mentioned in the first review were clarified in the methodology, discussion and by adding supplementary materials. The revised manuscript can be published in present form.
Reviewer 2 Report
The authors have answered and satisfactorily addressed all the reviewer's questions and concerns.